# Depressive symptoms and violence exposure in a population-based sample of adult women in South Africa

**Abigail M. Hatcher**[1,2]*, **Sthembiso Pollen Mkhize**[3], **Alexandra Parker**[3], **Julia de Kadt**[3]

**1** Gillings School of Global Public Health, University of North Carolina, Chapel Hill, North Carolina, United States of America, **2** School of Public Health, Faculty of Health Sciences, University of the Witwatersrand, Johannesburg, South Africa, **3** Gauteng City Regional Observatory, University of Johannesburg and University of the Witwatersrand, Johannesburg, South Africa

* abbeymae@email.unc.edu

**Data Availability Statement:** Public access data for use under a Creative Commons Attribution 4.0 International License available at https://www.datafirst.uct.ac.za/dataportal/index.php/collections/

## Abstract

Depressive symptoms are a major burden of disease globally and is associated with violence and poverty. However, much of the research linking these conditions is from resource-rich settings and among smaller, clinical samples. Secondary data from a household survey in Gauteng Province of South Africa examines the cross-sectional association between adult women's elevated depressive symptoms and markers of violence. Using tablet computers, participants self-completed interview modules to screen for depressive symptoms (Patient Health Questionnaire 2-item screener), childhood exposure to physical and sexual abuse (Childhood Trauma Questionnaire 4-item index), as well as past-year exposure to sexual or intimate partner violence (SIPV; WHO Multicountry Study instrument 4-item index). Socio-economic status, food security, education, and income were self-reported. Representative data at the ward level allows for modeling of results using survey commands and mixed-level modeling. Of the 7,276 adult women participating in the household survey, 42.1% reported elevated depressive symptoms. A total of 63.9% reported childhood violence exposure and 5.3% had past-year SIPV. Multi-level modeling suggests that violence is a strong predictor of depressive symptoms. Childhood abuse alone increases the odds of high depressive symptomology, after controlling for individual-level markers of poverty and neighborhood of residence (aOR 1.31, 95%, CI 1.17–1.37). Combined exposure to childhood abuse and past-year SIPV increased odds of reporting elevated depressive symptoms (aOR 2.05, 95%, CI 1.54–2.71). Ward characteristics account for 6% of the variance in depressive symptoms, over and above the contributions of household food security and socio-economic status. Exposure to violence in childhood and past-year SIPV were associated with depressive symptoms among women. These associations persist after controlling for socio-economic markers and latent neighborhood characteristics, which also had significant association with elevated depressive symptoms. These data suggest that efforts to reduce the burden of depressive symptoms may benefit from approaches that prevent violence against women and children.

GCRO or by requesting access to support@data1st.org.

**Funding:** The Quality of Life Survey 6 (2020/21) was funded by the Gauteng City-Region Observatory's core grant from the Gauteng Provincial Government, with additional financial contributions from the three metropolitan municipalities in Gauteng, South Africa. The funders had no role in study design, data collection and analysis, decision to publish, or preparation of the manuscript. SPM, AP and JDK received salaries from the Gauteng City-Region Observatory's core grant during the preparation of the article. AMH received payment as a consultant from the Gauteng City-Region Observatory's core grant for work on the preparation of this study.

**Competing interests:** The authors have declared that no competing interests exist.

## Introduction

Depressive symptoms represent a major mental health challenge for populations across the globe [1]. The Global Burden of Disease Study found that depressive disorders are among the most important causes of years lost to disability, especially among women [2], and this trend has only heightened since the start of the SARS-Cov2 pandemic [3]. In South Africa, national surveys suggest probable depression occurs among up to 28% of women [4], with considerable variation across the country [5].

Depressive symptoms among women in South Africa may be associated with their recent or childhood exposure to violence. Intimate partner violence (physical and/or sexual abuse by a current or ex-boyfriend [6]) is high in South Africa, with one population-based survey noting past-year exposure rates to sexual (8%) or physical (13%) violence [7]. Similarly sexual violence in the form of rape by a partner or non-partner is higher in South Africa than most settings globally, with 25–28% reporting ever experiencing rape [7, 8]. Childhood violence is similarly high in South Africa, with exposure to physical or sexual violence reported by 15% of girls in South Africa national population-based study [9].

Across cohort studies globally, there is robust evidence that violence exposure underpins mental health. A meta-analysis of cohorts found IPV exposure is associated with later elevated depressive symptoms [10]. Longitudinal South Africa studies drawn from clinical samples confirm this relationship between violence and later depression [11]. Childhood adversity can also have profound effects on mental health, with longitudinal research linking exposure to childhood abuse with anxiety and depression later in life [12, 13].

Poverty also frames the mental health of individuals in multiple ways. In South Africa, poverty is associated with worsened depressive symptoms [14, 15]. Poverty and depressive symptoms seem to be bi-directional, with new experimental evidence suggesting each condition worsens the other [16]. Poor mental health can also be a result of neighborhood conditions, since inadequate housing, physical degradation, or a lack of opportunity in the workforce are deeply intertwined with how people function psychologically [17–19]. An increasing number of studies from sub-Saharan Africa suggest that urban populations more frequently present with common mental disorders than their rural counterparts [20, 21]. This could be a result of the psychological toll of living in peri-urban settlements [22], neighborhood-level exposures to violence [23], or deprivation in terms of the living environment or employment prospects [24].

Overall, little population-based research from South Africa has measured the association of mental health and key structural determinants such as violence or socio-economic context. Filling this gap using stratified samples representative of wider populations can help programming and policy to target mental health and violence prevention policies in coming years.

## Methods

### Study setting

The Quality of Life survey has been conducted in Gauteng Province of South Africa every two years since 2009. In addition to being home to the major urban centers of Johannesburg and Pretoria, it comprises multiple burgeoning informal and peri-urban settlements. The province has a high degree of income inequality, and variable living conditions. In disadvantaged areas, quality of basic services (such as paved roads, sewerage, or electricity) is often poor. Public health, education and policing services are highly variable, and wealthier residents typically make use of private services [25].

 

## Study population and sampling

Quality of Life survey 6 interviewed 13 616 male and female adults across Gauteng, South Africa between the time of Sept 2019 –August 2020. The multistage stratified cluster sample design covered urban, peri-urban, and rural areas within the province and has been described fully elsewhere [26]. South African local government assigns neighborhoods as administrative wards. Within each (total of $n$ = 529 wards), 5 to 6 enumeration areas were sampled on a probability proportional to size (PPS) basis. PPS uses population data to assign a sampling probability variable so that between 4 to 6 dwelling units were randomly selected within each enumeration areas, using a randomly generated interval. A fixed sample size was set for each ward in the province (dependent on municipal location). Within the ward, each numeration area was assigned a sampling weight according to the number of dwelling unit, meaning the probability of selecting an enumeration area was proportional to its relative size in terms of number of dwelling units within the ward. A floor of 20 interviews was conducted in each ward. Data was weighted by population size, race and gender to ensure representative estimates.

## Procedures

On arrival at each dwelling unit, a trained enumerator spoke with residents to list all adults living in the dwelling. In-field random sampling selected one resident adult from a household roster for interview. This was achieved in practice by manually listing all resident adults and programming the data collection system (on tablet computers) to randomly select one adult to be interviewed. Participants were eligible if they were 18 years or older, willing to take part, and cognitively able to complete the interview.

Interviews were conducted in-person by trained enumerators following verbal informed consent. Participants were given a printed information sheet to keep, if they chose. All study materials were translated in nine local languages, and interviews were conducted in the language of choice.

An additional voluntary module was comprised of self-completed questions about experiences of violence. Self-completion can help protect participant confidentiality, ensures that answers can be completed without added distress of disclosure to another person, and can protect fieldworkers from vicarious trauma of listening to difficult stories. All participants were asked verbally if they chose to self-complete questions using a tablet programmed with questions in the language of their choice. Participants were given the option to skip this section, and the tablet locked before being returned to the fieldworker, so that no data around violence exposure was made known to any member of the study team.

## Measures

*Depressive symptoms* were assessed using the 2-item shortened version of the Patient Health Questionnaire (PHQ-2), which has been validated in South Africa [27]. This brief screening tool asks about frequency of two symptoms over the past two weeks (feeling down, depressed, and hopeless or losing pleasure or interest in doing things) with responses ranging on a Likert scale from 0 (not at all) to 3 (every day). PHQ-2 has been shown to have good sensitivity and adequate specificity across settings with a cut-point of 2+ suggesting probable depression [28]. While results of PHQ-2 are not clinically meaningful, they can provide a screening estimate of population-based prevalence of depressive symptoms.

*Childhood violence* exposure was asked using 4 items from the Childhood Trauma Questionnaire [29], a tool that has been validated in its entirety in South Africa [30]. For physical abuse, respondents were asked about whether they had been beaten with a belt, stick, or other

hard object, at home or at school, before the age of 18. To assess exposure to childhood sexual abuse, they were asked whether they had been molested (forced touching of genitals) or raped (forced sex or sex under threat), before the age of 18.

*Sexual or Intimate Partner Violence (SIPV)* was assessed using items from the WHO Multi-Country Study Instrument [31]. These items were selected from 10 possible items due to optimal discriminate value based on Item Response Theory in previous South African samples. SIPV includes past-year reports of any of the following by a current or former partner: hitting, kicking, threats or use of a weapon and/or forced sex. It also includes forced sex or sex under conditions of threat by a non-partner. The rationale for including this broader definition of SIPV is that rape in South Africa (by a partner or non-partner) is considered a pressing health and policy issue. Therefore, the survey wanted to establish baseline levels of exposure to both partner perpetration and non-partner perpetration.

*Socio-economic status* is a weighted measure of coverage by medical insurance, highest level of education completed, working internet access in the home, employment status, and household income, that ranges from 1–10 on a continuous scale. *Food insecurity* was measured using a non-validated measure of 5 items. The index comprised inadequate monthly expenditure on food relative to household size, an adult having skipped meals due to lack of money to buy food, and no place to purchase food within walking distance.

*Education* was assessed using the total number of years of schooling attained. This was dichotomized for descriptive statistics and bivariate analysis as achieving high school education or not. *Monthly income* was captured as categorical in the self-complete module. Participants were asked to include all money available to their household, including from work or social grants. These data were imputed using the R program MissForest [32] as missing values occurred in 1 963 instances (27% of sample). *Employment* was measured through a single item asking whether the participant worked (y/n) in the past 7 days. *Ward* is a latent construct based on groupings 20–32 participants living within a defined sub-region in the province.

## Quality control

As described fully elsewhere [33], data was quality was checked by quality assurance managers at the field level on a daily basis. Built-in checks were implemented through queries built into real-time data entry and in a back-end audit trail. Call backs were implemented to 26.6% of the full sample to check basic socio-demographics against the dataset. Any records that did not pass each of these three quality checks were omitted from the final dataset.

## Analysis

Descriptive statistics were assessed using STATA 16, using survey commands to adjust for clustering by ward and sampling strategies. To compare the sample characteristics of those participants who opted into the self-completed violence, section used chi-square statistics. Mixed-level regression used **melogit** commands, modeling neighborhood as a random effect. This technique allows neighborhood to serve as a latent confounding variable [34].

## Ethical considerations

Ethical approval was secured from the University of the Witwatersrand Human Research Ethics Committee (H19/11/09).

Dedicated training of field workers included activities on how to conduct the self-complete section, sensitivities around collecting information around violence, and how they could offer support if a participant wanted to speak to them about mental health or violence. A distress protocol helped to ensure that a basic level of calming containment was provided by field

workers, with the option to phone their field work team leader if additional support was required.

All participants, regardless of taking part in this portion of the interview or not, were given the option of taking a printed list of referrals for each provincial sub-region. The team phoned all listed referrals in advance to establish whether they were actively working and accepting new clients. All field team members were provided with three group debriefing sessions led by an experienced social worker. Field workers who struggled with vicarious trauma were additionally invited to take part in debriefing facilitated by a social worker who specialized in lay health worker training around violence and trauma.

## Results

### Descriptive statistics

A total of 7 276 adult females were included in the sample for this analysis. Women taking part ranged from age 18 to 86, with 13% being in the youngest age group (18–24 years). Weighted population-representative estimates suggest monthly household income was less than R 1600 monthly (about US$ 110) for one-third of women. Approximately half had high school education and more than two-thirds were currently unemployed. Over half of women were currently food insecure (Table 1).

The self-completed violence module was taken by 86% of women. Those women opting to take this module skewed younger, were more educated, and had higher income. They were more likely to report food insecurity (Table 1).

### Depressive symptoms and violence exposure

Of the entire sample of women participating in the survey, 42.1% reported elevated symptoms of depression as assessed by the PHQ-2 (Table 1).

The group taking part in the self-complete violence section had 42.7% reporting elevated depressive symptoms ($p$ = 0.013). Sexual abuse during childhood was reported by 12.9% of female participants (including 10.1% who reported molestation and 7.5% who reported rape). Physical abuse was reported by 63.1% of participants (including 47.1% who reported being beaten at home and 50.2% who were beaten at school).

**Table 1. Descriptive statistics of women in the survey (n = 7276).**

|  | 1 | 2 | 3 | 4 |
|---|---|---|---|---|
|  | Proportion in overall sample* (n = 7276) | Missing module (n = 993, 13.6%) | Completing module (n = 6283, 86.4%) | test for difference |
| Age <34 years | 13.0% | 16.8% | 38.3% | < .001 |
| Monthly income <R1600 | 30.3% | 31.5% | 23.1% | < .001 |
| High school education | 55.0% | 45.1% | 56.7% | < .001 |
| Unemployed | 68.4% | 70.0% | 68.1% | 0.259 |
| Food insecure | 56.6% | 43.7% | 58.8% | < .001 |
| Elevated depressive symptoms | 42.1% | 38.5% | 42.7% | 0.013 |
| Exposed to childhood violence | - | - | 63.9% | - |
| Past-year SIPV exposure | - | - | 5.3% | - |

aOR: adjusted odds ratio; SIPV: sexual or intimate partner violence

*Proprotions account for clustering by ward

Overall, 5.3% of female respondents reported past-year exposure to SIPV. This number includes women exposed to non-partner rape (*n* = 150, 1.7%) as well as those exposed to physical or sexual partner violence (*n* = 271, 4.4%). The two groups are not mutually exclusive as 35 participants experienced both non-partner rape and intimate partner violence.

## Bivariate associations

In unadjusted logistic regression, accounting for clustering by ward, multiple socio-economic markers were significantly associated with elevated depressive symptoms (Table 2). Lower income, no high school education, being unemployed in the past week, and food insecurity significantly increased odds of a woman reporting elevated depressive symptoms. Younger age was protective for depressive symptoms.

The reference group for examining odds of elevated depressive symptoms was women who reported no violence at any timepoint. Compared to these violence-free women, those reporting exposure to childhood violence alone had 37% greater odds of elevated depressive symptoms (*p* = <0.001). Past-year SIPV exposure was associated with 51% higher odds, but not at a significant level (*p* = 0.26). Women reporting both childhood abuse and past-year SIPV had nearly doubled odds of elevated depressive symptoms (*p* = <0.001).

## Mixed effects model

Mixed effects modeling showed strong associations between current elevated depressive symptoms, violence exposure, and poverty (Table 3). Compared to women reporting no violence, those with childhood violence had 31% higher odds of adult depressive symptoms (aOR 1.31, 95%, CI 1.17–1.37). The small number of women reporting past-year SIPV but never reporting childhood abuse also had higher odds of depressive symptoms, but this was not statistically significant (aOR 1.31, 95%, CI 0.72–2.39). Those reporting both childhood abuse and past-year SIPV had more than doubled odds of high depressive symptoms (aOR 2.05, 95%, CI 1.54–2.71).

In the final model, food insecurity heightened reports of depressive symptoms, whether this was moderate (28% increased odds) or severe (83% increased odds). Social class as a weighted index was predictive of depressive symptoms, with each 1-point increase in the scale associated with 9% lower odds of reporting depressive symptoms. Intra-class correlation suggests that 6% of the variance in depressive symptoms can be attributed to the ward of residence.

**Table 2. Associations of key covariates and elevated depressive symptoms.**

|  | OR | 95% CI | p value |
|---|---|---|---|
| Age <34 years | 0.79 | (0.70–0.90) | <0.001 |
| Monthly income <R1600 | 1.46 | (1.28–1.66) | <0.001 |
| Lacks high school education | 1.52 | (1.35–1.72) | <0.001 |
| Unemployed | 1.41 | (1.24–1.62) | <0.001 |
| Food insecure | 1.72 | (1.51–1.95) | <0.001 |
| No violence ever | Ref | - - | - |
| Exposed to childhood abuse | 1.37 | (1.19–1.57) | <0.001 |
| Past-year SIPV exposure | 1.51 | (0.74–3.08) | 0.26 |
| Both childhood abuse & past year violence | 1.92 | (1.36–2.72) | <0.001 |

OR: odds ratio; SIPV: sexual or intimate partner violence

Model accounts for clustering by ward

**Table 3. Mixed effects model of elevated depressive symptoms in a stratified cluster sample of women (*n* = 6093).**

|  | aOR | 95% CI | p value |
|---|---|---|---|
| No violence ever | Ref |  |  |
| Childhood abuse | 1.31 | (1.17–1.47) | <0.001 |
| Past-year SIPV exposure | 1.31 | (0.72–2.39) | 0.375 |
| Both childhood & SIPV | 2.05 | (1.54–2.71) | <0.001 |
| Age | 1.06 | (1.04–1.08) | <0.001 |
| Food secure |  |  |  |
| Moderate food insecurity | 1.28 | (1.13–1.45) | <0.001 |
| Severe food insecurity | 1.83 | (1.49–2.24) | <0.001 |
| Social class | 0.91 | (0.89–0.93) | <0.001 |
| ICC variance by Ward | 0.06 | (0.04–0.08) | <0.001 |

aOR: adjusted odds ratio; SIPV: sexual or intimate partner violence; ICC: intra-class correlation

Model accounts for clustering by ward

Results are qualitatively similar in a sensitivity analysis using 3 as the cut-point for the PHQ-2 (Table A in S1 Appendix). A sensitivity analysis explored assumptions about the 13.6% who chose not to self-complete the violence module. If these participants are assumed to be violence-free or if they are assumed to be exposed to one or more forms of violence, the key associations between violence exposure and elevated depressive symptoms hold (Tables B and C in S1 Appendix).

## Discussion

High rates of elevated symptoms of depression were found in a population-based sample of women in urban and peri-urban South Africa. After controlling for latent neighborhood characteristics, age, food security, and social class, exposure to violence was independently associated with poor mental health. Combined exposure to childhood abuse and SIPV more than doubled the odds of women reporting high depressive symptoms. Childhood abuse on its own was independently associated with adult depressive symptoms.

Population-based estimates of violence exposure add to a limited body of research in South Africa. The estimate of 12.9% reporting childhood sexual violence are similar to a national population-based study finding that 14.6% of girls in South Africa report childhood sexual abuse [9]. The 4.4% estimate of past-year IPV is lower than 13.2% found in a similar study by Machisa *et al.*, who conducted a household study in Gauteng Province [7].

On the other hand, our estimate of past-year rape is more than 20-fold higher than the most recent police statistics, which suggest that in 2021, 0.069% of the Gauteng population reported past-year rape to the police [35]. This paper is among the first population-based estimates of rape incidence in a high-risk region of South Africa.

Latent characteristics of one's ward of residence seem to explain considerable variation in depressive symptoms, over and above individual social class and food security. There seem to be multiple plausible reasons for neighborhood residence and depressive symptoms. The physical infrastructure of a community can be linked to depressive symptoms by causing increased daily stressors and strain [22, 36]. Crime and witnessing violence can directly impact on mental health [23, 37]. It is also possible that those who live in poor socio-economic areas have fewer social or institutional resources to cope with strains or threats to safety when exposed [19, 24].

The distinction between the study sample completing the self-report violence section and those declining warrants attention. A relatively younger sample may have reported more depressive symptoms, since violence and depression are found at higher rates in younger women compared to older peers [38–40]. Those taking part were more food insecure than those declining, which may have altered the models given strong links from hunger to depression [41–43].

These sample considerations should be viewed in light of other study limitations. Cross-sectional data can only suggest associations, rather than causality. The brief PHQ-2 screener cannot be used to diagnose clinical depression. The non-validated measure of food insecurity and the continuous additive scale for social class has limitations compared to existing measures or principal component analysis, but these poverty markers needed to align with past survey rounds in order to track trends over time. A brief violence measure was used within a broader survey, which limits sensitivity to detecting cases of violence exposure–a more robust set of items would have likely identified higher incidence of SIPV. The measure of childhood exposure to violence does not provide detail on duration or source of abuse, but as shown by other studies, some reported childhood violence may have been enacted by a partner [44, 45].

Scholars have noted that studies dedicated to violence–and for whom staff training and support centers around skills of safely asking and hearing about disclosures of violence–have more accurate findings than general surveys where a violence module is added [46]. This leads to consideration of these SIPV estimates, in particular, as a minimum level of population-based exposure. It is unclear whether higher estimates of SIPV exposure would alter the relationship between exposure and high depressive symptoms.

## Conclusion

Exposure to violence was independently associated with high depressive symptoms among a relatively large, population-based sample of women in South Africa. These data may contribute to national efforts to reduce violence against women and girls. Global efforts to improve mental health could gain traction by supporting individuals who are exposed to high rates of violence and increasing resources to communities where poverty and safety intersect. Preventing violence within communities represents a crucial step for ensuring health and well-being for women and children globally.

## Supporting information

**S1 Appendix.** Table A: Mixed effects model of elevated depressive symptoms at cutpoint of PHQ >3 in a stratified cluster sample of women (n = 6093). Table B: Mixed effects model of elevated depressive symptoms assuming non-completers are violence-free (n = 6093). Table C: Mixed effects model of elevated depressive symptoms assuming non-completers are violence-exposed (n = 6093).
(DOCX)

## Acknowledgments

We acknowledge GeoSpace and the team of trained enumerators, our colleagues at GCRO and local government, and the many participants who took part in this survey.

## Author Contributions

**Conceptualization:** Abigail M. Hatcher, Sthembiso Pollen Mkhize, Alexandra Parker, Julia de Kadt.

**Data curation:** Sthembiso Pollen Mkhize, Alexandra Parker, Julia de Kadt.

**Formal analysis:** Abigail M. Hatcher, Alexandra Parker, Julia de Kadt.

**Funding acquisition:** Julia de Kadt.

**Investigation:** Abigail M. Hatcher, Sthembiso Pollen Mkhize, Alexandra Parker, Julia de Kadt.

**Methodology:** Abigail M. Hatcher, Sthembiso Pollen Mkhize, Julia de Kadt.

**Project administration:** Sthembiso Pollen Mkhize, Alexandra Parker, Julia de Kadt.

**Resources:** Julia de Kadt.

**Software:** Sthembiso Pollen Mkhize.

**Supervision:** Abigail M. Hatcher, Sthembiso Pollen Mkhize.

**Validation:** Abigail M. Hatcher, Sthembiso Pollen Mkhize, Julia de Kadt.

**Visualization:** Alexandra Parker.

**Writing – original draft:** Abigail M. Hatcher.

**Writing – review & editing:** Abigail M. Hatcher, Sthembiso Pollen Mkhize, Alexandra Parker, Julia de Kadt.

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
