## [Decision Letter · Decision Letter 0]

22 Apr 2022

PGPH-D-22-00209

Depressive symptoms and violence exposure in a population-based sample of adult women in South Africa

Dear Dr. Hatcher,

Thank you for submitting your manuscript to PLOS Global Public Health. After careful consideration, we feel that it has merit but does not fully meet PLOS Global Public Health’s publication criteria as it currently stands. Therefore, we invite you to submit a revised version of the manuscript that addresses the points raised during the review process.

Please submit your revised manuscript by . If you will need more time than this to complete your revisions, please reply to this message or contact the journal office at globalpubhealth@plos.org. Please include the following items when submitting your revised manuscript:

We look forward to receiving your revised manuscript.

Kind regards,

Ahmed Waqas

Academic Editor

Journal Requirements:

1. Please send a completed 'Competing Interests' statement, including any COIs declared by your co-authors. If you have no competing interests to declare, please state "The authors have declared that no competing interests exist".

3. We notice that your supplementary tables are included in the manuscript file. Please remove them and upload them with the file type 'Supporting Information'. Please ensure that each Supporting Information file has a legend listed in the manuscript after the references list.

4. Please note that your Data Availability Statement is currently missing a direct link to access each database. If your manuscript is accepted for publication, you will be asked to provide these details on a very short timeline. We therefore suggest that you provide this information now, though we will not hold up the peer review process if you are unable.

Additional Editor Comments (if provided):

Dear Dr. Hatcher,

Thank you for considering PloS Global Public Health for submission of your manuscript. The manuscript is well-written and explores an important public health topic. I have received a favorable response from both the reviewers. Could you please revise your manuscript in line with their comments? I look forward to reading the revised manuscript.

Best wishes,

Dr. Ahmed Waqas

Reviewers' comments:

Reviewer's Responses to Questions

**Comments to the Author**

1. Does this manuscript meet PLOS Global Public Health’s publication criteria? Is the manuscript technically sound, and do the data support the conclusions? The manuscript must describe methodologically and ethically rigorous research with conclusions that are appropriately drawn based on the data presented.

Reviewer #1: Partly

Reviewer #2: Yes

2. Has the statistical analysis been performed appropriately and rigorously?

Reviewer #1: Yes

Reviewer #2: Yes

3. Have the authors made all data underlying the findings in their manuscript fully available (please refer to the Data Availability Statement at the start of the manuscript PDF file)?

Reviewer #1: Yes

Reviewer #2: No

4. Is the manuscript presented in an intelligible fashion and written in standard English?

Reviewer #1: Yes

Reviewer #2: Yes

5. Review Comments to the Author

Reviewer #1: I want to appreciate the authors for their time and effort in doing this very important public health issue. The following comment needs to be addressed for the additional value of the paper. The paper needs to be revised the grammar, editing (you used present illness), list acronyms/ abbreviations (when you used in the first line for example line 110, PHQ), and avoid using words such as our, us, and we throughout the paper.

The Abstract section is good but the conclusion is too general, in this you also said that "Individual markers of poverty" and "Reduction of poverty" what do you mean. Your conclusion is general it should be specific and based on your finding. In the Introduction section, Paragraph two (line 62) the way of citation need to be corrected and what does elsewhere mean? In general, the introduction is good, but you failed to address, depression and violence from global to local and what efforts was done in South Africa, consequence, magnitude, and severity of violence and depression are not clearly addressed in this section.

In the method section, it is good but there is no clear information about the study area, study time, study design, population (source and study population/inclusion and exclusion criteria), sample size determination, sampling technique, and data quality control separately (subtitle) in this paper. The measurement (operational definitions) is good but the way that you measured some variables is somehow confused. Line 119 you said that "These 4 items have not been validated", in this, the definition of an intimate partner is defined as also for non-partner (why?), how did you measure this? Also, intimate partner violence is broad and includes physical, sexual, and psychological violence but you only used some of them? The socioeconomic status and food insecurity measurement are also not clear (why you did not use Principal component analysis for the economic status?) In the ethical considerations, you included the funding issue, it is better to list in the funding report.

In the Result section, avoid words that are not scientific such as Just (line 161). Avoid (columns 2-4) in line 167 when you referred to table 1. It would be better for readers if you show the results of depressed symptoms and violence in a separate paragraph (make in one paragraph from lines 170-179) b/ce it is about violence. Line 183, change the word "lacking high school education"

Discussion: In this, some parts of the justification are cited, it be better to justify the possible reasons for both the magnitudes and for factors (for example lines 227-236). Line 242 what does theoretically mean? The conclusion did not show the finding and not concluded based on your pertinent findings in addition to that you cited the reference (42) while it is unnecessary. Conclude your finding based on your objective. There are no abbreviation, author contribution, funding information, ethical issue, data availability, Acknowledgement, consent and conflict of interest statement in this paper. Follow the guideline of the journal for tables and edit some of the references and avoid reference number 42 from the citation (conclusion) and reference list.

Reviewer #2: Overall, this is a well-written manuscript. The authors have systematically examined the topic of investigation.

There are a few specific comments as mentioned below.

1. Line 34-38: Abstract: - Mention Odds Ratio and 95% confidence interval for significant associations.

2. Line 82-86: Elaborate on sampling method used. How many total wards and Enumeration Areas are there in the province? Do they cover rural, urban or both areas? What proportion was chosen for PPS sampling? Which type of random sampling was used to select dwelling units? were chosen out of how many total wards available? If more than one adult women were there in a dwelling, how were they selected?

3. Line 106-107, 114-115, 120-121: Were the data collection tools translated to local language? Mention it clearly.

4. Line 138-139: Description of a ward should be part of methods section where the study area is mentioned. Move it there.

5. Line 188-192: Provide Odds Ratio and 95% confidence interval in bracket for significant associations.

6. PLOS authors have the option to publish the peer review history of their article (what does this mean?). If published, this will include your full peer review and any attached files.

**Do you want your identity to be public for this peer review?** For information about this choice, including consent withdrawal, please see our Privacy Policy.

Reviewer #1: No

Reviewer #2: **Yes: **DEEPANJALI BEHERA

---

## [Decision Letter · Decision Letter 1]

22 Jul 2022

PGPH-D-22-00209R1

Depressive symptoms and violence exposure in a population-based sample of adult women in South Africa

Dear Dr. Hatcher,

Thank you for submitting your manuscript to PLOS Global Public Health. After careful consideration, we feel that it has merit but does not fully meet PLOS Global Public Health’s publication criteria as it currently stands. Therefore, we invite you to submit a revised version of the manuscript that addresses the points raised during the review process.

We look forward to receiving your revised manuscript.

Kind regards,

Ahmed Waqas

Academic Editor

Journal Requirements:

1. In the Funding Information you indicated that no funding was received. Please revise the Funding Information field to reflect funding received.

Please ensure that the funders and grant numbers match between the Financial Disclosure field and the Funding Information tab in your submission form. Note that the funders must be provided in the same order in both places as well.

2. We have noticed that you have uploaded Supporting Information files, but you have not included a list of legends. Please add a full list of legends for your Supporting Information files after the references list.

Additional Editor Comments (if provided):

Reviewers' comments:

Reviewer's Responses to Questions

**Comments to the Author**

1. If the authors have adequately addressed your comments raised in a previous round of review and you feel that this manuscript is now acceptable for publication, you may indicate that here to bypass the “Comments to the Author” section, enter your conflict of interest statement in the “Confidential to Editor” section, and submit your "Accept" recommendation.

Reviewer #1: All comments have been addressed

Reviewer #2: (No Response)

2. Does this manuscript meet PLOS Global Public Health’s publication criteria? Is the manuscript technically sound, and do the data support the conclusions? The manuscript must describe methodologically and ethically rigorous research with conclusions that are appropriately drawn based on the data presented.

Reviewer #1: Yes

Reviewer #2: Yes

3. Has the statistical analysis been performed appropriately and rigorously?

Reviewer #1: Yes

Reviewer #2: Yes

4. Have the authors made all data underlying the findings in their manuscript fully available (please refer to the Data Availability Statement at the start of the manuscript PDF file)?

Reviewer #1: Yes

Reviewer #2: No

5. Is the manuscript presented in an intelligible fashion and written in standard English?

Reviewer #1: Yes

Reviewer #2: Yes

6. Review Comments to the Author

Reviewer #1: Majority of the comments are addressed. But there are some editorial errors that still need to be addressed throughout the paper. Subjective words such as our, us are also still common in the paper that needs to be corrected. The conclusion part is still not corrected. No need of citation in the conclusion part and you should conclude based on your pertinent finding and avoid citation.

Reviewer #2: There is a good improvement in the revised version.

1. There is still a need to make the sampling description more clear and specific. What proportion was chosen for PPS sampling? Which type of random sampling was used to select dwelling units? How was the total sample size reached at by combining respondents selected at all levels? If more than one adult respondents were available in a dwelling, how were they selected?

2. In Line 113, it is mentioned that 13616 interviews were carried out. But, the same number is regarded as the overall study population in Line 219. It also states that 7276 women were included as the sample. Were there a total of 7276 women in the database of 13616 respondents which was used as part of this study? If they were more, how were these 7276 women chosen? There is a need to present these details with more clarity.

7. PLOS authors have the option to publish the peer review history of their article (what does this mean?). If published, this will include your full peer review and any attached files.

**Do you want your identity to be public for this peer review?** For information about this choice, including consent withdrawal, please see our Privacy Policy.

Reviewer #1: No

Reviewer #2: **Yes: **Deepanjali Behera

---

## [Editor Report · Decision Letter 2]

20 Sep 2022

Depressive symptoms and violence exposure in a population-based sample of adult women in South Africa

PGPH-D-22-00209R2

Dear Dr Hatcher,

We are pleased to inform you that your manuscript 'Depressive symptoms and violence exposure in a population-based sample of adult women in South Africa' has been provisionally accepted for publication in PLOS Global Public Health.

Before your manuscript can be formally accepted you will need to complete some formatting changes, which you will receive in a follow up email. A member of our team will be in touch with a set of request. Please note that your manuscript will not be scheduled for publication until you have made the required changes, so a swift response is appreciated.

Best regards,

Ahmed Waqas

Academic Editor